# Simulated Microgravity-Induced Alterations in PDAC Cells: A Potential Role for Trichostatin A in Restoring Cellular Phenotype

**DOI:** 10.3390/ijms26104758

**Published:** 2025-05-16

**Authors:** Corinna Anais Pagano, Maria Angela Masini, Maurizio Sabbatini, Giorgia Gribaudo, Marcello Manfredi, Flavia Giusy Caprì, Valentina Bonetto, Valeria Magnelli, Massimo Donadelli, Roberto Corino, Masho Hilawie Belay, Elisa Robotti, Emilio Marengo

**Affiliations:** 1Department of Science and Innovation Technology (DISIT), Università del Piemonte Orientale, 15121 Alessandria, Italy; corinnaanais.pagano@uniupo.it (C.A.P.); maria.masini@uniupo.it (M.A.M.); 20033773@studenti.uniupo.it (G.G.); flavia.capri@uniupo.it (F.G.C.); valentina.bonetto@uniupo.it (V.B.); valeria.magnelli@uniupo.it (V.M.); masho.belay@uniupo.it (M.H.B.); elisa.robotti@uniupo.it (E.R.); emilio.marengo@uniupo.it (E.M.); 2Department of Translational Medicine (DIMET), University of Eastern Piedmont, 28100 Novara, Italy; marcello.manfredi@uniupo.it; 3Institute for Molecular and Translational Cardiology (IMTC), IRCCS Policlinico San Donato, 20097 Milan, Italy; 4Department of Neurosciences, Biomedicine and Movement Sciences (DNBM), University of Verona, 37134 Verona, Italy; massimo.donadelli@univr.it; 5Centro Italiano di Diagnostica Medica Ultrasonica (CIDIMU), 10128 Turin, Italy; corino.gymnasium@gmail.com

**Keywords:** spheroid, 3D culture, proteomic analysis, microgravity, random position machine, bioinformatics, pathway analysis

## Abstract

Pancreatic ductal adenocarcinoma (PDAC) accounts for 90% of all pancreatic malignancies. Despite the remarkable improvement concerning treatment, late detection and resistance to clinically used chemotherapeutic agents remain major challenges. Trichostatin A (TSA), a histone deacetylase inhibitor, has been recognized as an effective therapeutic agent against PDAC by inhibiting proliferation, inducing apoptosis, and sensitizing PDAC cells to chemotherapeutic agents such as gemcitabine. Microgravity has become a useful tool in cancer research due to its effects on various cellular processes. This paper presents a deep molecular and proteomic analysis investigating cell growth, the modulation of cytokeratins, and proteins related to apoptosis, cellular metabolism, and protein synthesis after TSA treatment in simulated microgravity (SMG)-exposed PaCa44 3D cells. Our analysis concerns the effects of TSA treatment on cell proliferation: the impairment of the cell cycle with the downregulation of proteins involved in Cdc42 signaling and G1/G2- and G2/M-phase transitions. Thus, we observed modification of survival pathways and proteins related to autophagy and apoptosis. We also observed changes in proteins involved in the regulation of transcription and the repair of damaged DNA. TSA treatment promotes the downregulation of some markers involved in the maintenance of the potency of stem cells, while it upregulates proteins involved in the induction and modulation of the differentiation process. Our data suggest that TSA treatment restores the cell phenotype prior to simulated microgravity exposure, and exerts an intriguing activity on PDAC cells by reducing proliferation and inducing cell death via multiple pathways.

## 1. Introduction

The progression of cancer is driven by a complex interplay of genetic mutations and epigenetic modifications. While the activation of oncogenes and inactivation of tumor suppressor genes are central, epigenetic alterations, such as DNA methylation and histone acetylation, have emerged as critical regulators of gene expression and promising therapeutic targets due to their reversibility [1]. Histone acetylation, typically associated with gene activation through the loosening of histone–DNA interactions, contrasts with histone deacetylation, which leads to gene silencing. Pharmacological manipulation of these processes has led to the development of histone deacetylase (HDAC) inhibitors, aiming to restore the balance of gene expression in cancer cells [2,3,4,5]. HDAC inhibitors, including Trichostatin A (TSA), have demonstrated preclinical potential in reversing epigenetic silencing and reactivating tumor-suppressive pathways [6]. The tumor suppressor gene TP53, a key player in cellular stress responses, is frequently mutated in pancreatic ductal adenocarcinoma (PDAC), one of the most lethal cancers [7]. Mutant p53 often gains new oncogenic functions, contributing to PDAC’s resistance to standard therapies like gemcitabine and 5-fluorouracil, as well as its aggressive characteristics, including early metastasis and chemoresistance [8,9]. The frequent absence of functional p53 in PDAC necessitates the exploration of alternative therapeutic strategies. HDAC inhibitors have emerged as promising candidates in this context, as they can induce the expression of cell cycle inhibitors such as p21WAF1/CIP1 through p53-independent mechanisms [10]. By inhibiting cyclin-dependent kinases, p21 can arrest cell proliferation, providing a rationale for the use of HDAC inhibitors like TSA in PDAC to potentially overcome limitations imposed by p53 mutations [11,12,13,14,15]. Furthermore, cancer resistance to therapeutic agents is significantly influenced by the tumor microenvironment, encompassing cell–cell and cell–matrix interactions, as well as the complex three-dimensional (3D) architecture of solid tumors [16]. These factors enable cancer cells to evade apoptotic signals and resist drug cytotoxicity [17]. Traditional two-dimensional (2D) cell culture models, widely used in preclinical testing, fail to adequately replicate the in vivo tumor microenvironment, lacking critical aspects like nutrient gradients, hypoxic regions, and variations in cell proliferation [18]. This limitation has spurred the development and utilization of 3D culture systems, which better mimic the physiological conditions of solid tumors and allow for a more accurate evaluation of drug efficacy [19,20,21]. Three-dimensional models, such as multicellular tumor spheroids, offer significant advantages by reproducing the structural and functional heterogeneity, extracellular matrix production, and hypoxic cores characteristic of in vivo tumors. These features are crucial for studying drug penetration and the metabolic adaptations of cancer cells [22]. Moreover, 3D cultures provide a platform to investigate drug resistance mechanisms specific to the tumor microenvironment, offering insights into potential therapeutic targets [23,24,25]. The random positioning machine (RPM), originally designed for gravitational biology and space medicine, has become an innovative tool in cancer research [26]. The RPM simulates microgravity, inducing cellular detachment and the formation of 3D tumor spheroids that closely mimic the metastatic behavior of cancer cells, providing a unique in vitro metastasis model [27]. Its ability to induce spheroid formation and promote molecular changes similar to those observed in metastasis has broadened its application in oncological studies [28]. The use of the RPM underscores the importance of dynamic culture systems that better replicate the physical and biochemical conditions of tumor progression. By facilitating the development of 3D tumor models, the RPM allows researchers to study cellular behaviors like migration, invasion, and drug resistance under conditions that simulate the in vivo setting [29,30,31]. The integrated study of epigenetic regulation and the tumor microenvironment is revolutionizing our understanding of cancer biology and treatment. HDAC inhibitors, like TSA, offer a promising strategy for targeting tumors with p53 mutations, such as PDAC, by utilizing alternative pathways to induce cell cycle arrest and apoptosis [11]. However, the inherent limitations of 2D cell culture models in accurately evaluating drug efficacy necessitate the adoption of advanced 3D culture systems [23]. Tools like the RPM further enhance our ability to study the metastatic and resistant behaviors of cancer cells, paving the way for the development of more effective therapeutic strategies [27]. By integrating these novel approaches, the field is advancing towards more predictive and clinically relevant models of cancer, ultimately aiming to improve patient outcomes [32]. This research paper presents a comprehensive molecular and proteomic investigation into the effects of Trichostatin A (TSA) on pancreatic adenocarcinoma PaCa44 cells cultured in a three-dimensional (3D)-simulated microgravity (SMG) also known as random positioning environment. The study specifically examines how TSA influences cell growth, the modulation of cytokeratins, and proteins associated with apoptosis, cellular metabolism, and protein synthesis. By analyzing these critical cellular processes, the researchers aim to elucidate the mechanisms through which TSA impacts cancer cell behavior under these unique conditions. The application of deep molecular and proteomic techniques is intended to provide a thorough understanding of TSA’s effects, potentially revealing new therapeutic targets and insights into its ability to inhibit cancer growth within a random positioning context, a condition particularly relevant due to the known influence of microgravity on cancer cell behavior and treatment effectiveness [33,34].

## 2. Results

### 2.1. Differential Effects of Trichostatin A on 2D and 3D PaCa44 Cell Cultures at 1 g

To evaluate the effects of Trichostatin A on pancreatic ductal adenocarcinoma (PDAC), PaCa44 cells were cultured under two distinct conditions: 2D monolayer culture and 3D culture. The cells were then treated with different concentrations of Trichostatin A to assess its cytotoxicity and identify the optimal dosage required to effectively induce apoptosis in PDAC cells.

The dose–response curves, as represented in Figure 1, summarize the effects of Trichostatin A on the viability of PaCa44 cells cultured under different conditions. In the 2D culture model, there is a quite pronounced dose-dependent reduction in viability exhibited by PaCa44 cells with increasing TSA concentrations. In contrast, the response to TSA was considerably blunted in the 3D spheroid model (Figure 1B); the dose–response curve was significantly flatter, indicative of a reduced sensitivity to the drug.

### 2.2. Proteomic Pathway Modulations Induced by Trichostatin A Treatment Under Simulated Microgravity in Pancreatic Cancer Spheroids

To investigate the biological alterations induced by TSA after simulated microgravity (SMG) exposure, PaCa44 cells were cultured in RPM culture conditions (as detailed in Section 4). After 9 days of exposure to simulated microgravity, including the last 48 h with TSA 2.5 μM treatment, an extended proteomic analysis was performed with SWATH-MS using a statistical cut-off of *p*-value < 0.05 and fold change > 1.3. The results showed that a 48 h treatment with TSA 2.5 μM and a total of 9 days of exposure to SMG induced the modulation of 349 proteins (115 up- and 234 downregulated, compared to control culture conditions, PaCa44 cells exposed to SMG for 9 days). The proteomic data set, including UniProt code and fold changes, was then uploaded to the Ingenuity Pathway Analysis (IPA) bioinformatics tool, to identify the canonical pathways significantly modulated. After 48 h of TSA 2.5 μM treatment and 9 days of SMG exposure more than one hundred pathways were altered, including 4 predicted activated (z-score > 2) and 31 predicted inhibited (z-score < −2; Figure 2B). Evaluating altered pathways identified by the proteomic analysis, we discovered that 48 h of TSA 2.5 μM treatment during the 9 days of SMG exposure inhibited two main pathways involved in cell proliferation and migration of tumoral cells (Figure 2C,D). In detail, we observed a downregulation of proteins involved in the G1/S and G2/M checkpoint regulation (PA2G4, SFN, YWHAG, and YWHAQ). Proteins like RHOA, Cdc42, ACTN4, EZR, MYH9, FLNB, and FSCN1 all involved in regulating the actin cytoskeleton and crucial for cellular adhesion and shape are downregulated as well. Moreover, a downregulation has been observed for ITGB1, LGALS3, SDCBP, PODXL, and CDCP1 proteins which are involved in cell–cell and cell–matrix adhesion. Moreover, several proteins involved in stress response (HSPA1B, HSPA5, and SERPINH1) and metabolism were detected. These results clearly underlie the anti-proliferative effects of TSA 2.5 μM on PaCa44 cells.

### 2.3. Disruption of EMT and Cytoskeletal Dynamics by Treatment: Impairment of Actin-Based Motility and Invasive Potential in Cancer Cells

To validate the effect of TSA 2.5 μM treatment on cell migration following simulated microgravity (SMG), we analyzed EMT markers and cytoskeletal dynamics in PaCa44 cells. EMT, a complex biological process where epithelial cells lose cell–cell junctions and polarity, and gain migratory and invasive properties [35,36,37], was a key focus in our analysis. RT-qPCR showed no change in E-cadherin (CDH1) mRNA, but a downregulation of Zeb1 mRNA, indicating a potential suppression of mesenchymal traits (Figure 3A). Western blot analysis revealed upregulated E-cadherin protein, with no differential regulation of Vimentin protein (Figure 3B). Proteomic analysis of cells exposed to 9 days of SMG with 48 h of TSA 2.5 μM treatment showed a downregulation of protein pathways related to regulation of actin-based mobility by Rho. Specifically, we observed a downregulation of several proteins involved in cytoskeletal dynamics (ACTB, ACTC1, ACTR2), actin regulators (CFL1, PFN1, ARPC5), and Rho GTPases (CDC42, RHOA, ITGB1) [38,39,40]. An upregulation of RHOC and HLA-DRB1 was also noted (Figure 3D). Furthermore, we evaluated the expression of cytoskeletal components. Western blot analysis showed a downregulation of α-tubulin and vinculin, while β-actin expression remained unchanged in the TSA-treated group compared to the control (Figure 3C). The proteomic analysis also revealed a significant downregulation of proteins associated with Cdc42 signaling (ACTR2, ARPC5, CDC42, CFL1, HLA-A, HLA-B, HLA-DRA, ITGB1, MYL6, PPP1CB). Collectively, these results suggest that the treatment disrupts EMT and cytoskeletal organization, likely impairing the migratory and invasive capabilities of cancer cells [41,42].

### 2.4. Impact of TSA (2.5 μM) on Apoptosis, PI3K/AKT, ERK, IL-8, HIF-1α Signaling, and Stemness Markers

TSA treatment significantly alters apoptosis- and stemness-related pathways. Apoptosis-related gene expression analysis revealed increased mRNA levels for caspase 3 and 9, alongside decreased caspase 8 mRNA, and a significant increase in BAD mRNA. Protein analysis indicated reduced activity in the PI3k/Akt and HIF-1α pathways, affecting glucose use, apoptosis, and cell growth. Specifically, proteins involved in these pathways, including YWHAB, YWHAH, YWHAZ, YWHAE, YWHAG, PPP2R1A, HSP90β, RAP1A, ITGB1, HSPA5, HSPA8, MMP14, SLC2A1, LDHA, RACK1, and PKM, were downregulated. Lower levels of HSPA5 and HSPA8 could disrupt protein folding and the cell’s response to stress in the endoplasmic reticulum, potentially leading to cell death. Reduced RAP1A and MMP14 could decrease cell movement and invasion. Changes in glucose metabolism might occur due to lower SLC2A1 and LDHA, potentially affecting energy production and cell growth [43,44]. Reduced RACK1 could disrupt various signaling pathways important for cell growth, survival, and specialization. Finally, lower PKM could impair the breakdown of glucose for energy [43,44]. TSA treatment also diminishes stem-like characteristics. Protein analysis showed decreased activity in the ERK5 and IL-8 signaling pathways. IL-8 is involved in stemness, new blood vessel formation, and EMT [45]. Lower levels of ITGB1, PA2G4, S100A4, ALDH1A3, and EpCAM, proteins associated with cell growth and stem cell markers, were observed. While these decreases could inhibit tumor growth and spread, HMGA1, a protein involved in cell proliferation and gene regulation, was upregulated (Figure 4B). Western blot analysis showed decreased CD44 levels, but no change in NANOG levels (Figure 4C). NANOG is a key regulator of stem cell properties, and CD44 is involved in cell adhesion, movement, and signaling, both playing roles in cancer stem cells (CSCs) in PDAC [46,47].

Finally, TSA treatment affects glucose metabolism and cell–cell junctions. RT-qPCR analysis showed a significant decrease in GAPDH mRNA (Figure 5A), while Western blot analysis showed no change in GAPDH protein levels (Figure 5B). The remodeling of epithelial adherens junctions signaling pathway was downregulated, with decreased levels of proteins like RAB5C, VCL, TUBA4A, ACTN4, DNM2, ARPC5, ACTR2, ACTC1, RAB7A, TUBB6, TUBB4B, TUBA1B, ACTB, and TUBB, and an upregulation of CTNNA1 (Figure 5C). The decrease in tubulin- and actin-related proteins would likely disrupt the cell’s internal scaffolding, impair transport, weaken cell–cell connections, and interfere with cell division, impacting cell movement, migration, and overall cell function in PDAC cells [48,49]. CTNNA1 upregulation might strengthen cell–cell connections and alter cell communication [50].

## 3. Discussion

The present study investigated the effects of Trichostatin A (TSA) on pancreatic ductal adenocarcinoma (PDAC) cells cultured in 2D and 3D, followed by proteomic analysis to identify the molecular mechanisms of TSA treatment under simulated microgravity (SMG). We observed that PaCa44 cells exhibited different sensitivities to TSA depending on the culture conditions. TSA showed significant cytotoxicity in 2D cultures but was less effective in 3D spheroids. This highlights the importance of the cellular microenvironment in assessing drug efficacy, as 3D models better mimic in vivo tumor conditions [51]. Indeed, spheroid cultures have become a valuable tool in cancer research for their ability to recapitulate key aspects of the tumor microenvironment [52,53,54]. Previous research by Masini et al. (2022) [31] demonstrated that prolonged (9-day) SMG exposure induces significant changes in PDAC cell growth, proliferation, and metabolic pathways, also influencing metastatic potential. Notably, simulated microgravity has been shown to alter the behavior of cancer cells and enhance their metastatic potential [33]. TSA, a histone deacetylase inhibitor, regulates gene expression and cell cycle progression by modulating histone acetylation [55]. This epigenetic regulation is crucial in understanding how microgravity affects cancer biology, potentially leading to new therapeutic strategies against chemotherapy resistance and metastasis. Combining epigenetic modulators like TSA with conventional chemotherapy could enhance treatment efficacy by resensitizing cancer cells [56]. Our SWATH-MS analysis, a powerful and advanced proteomic technology that offers reliable and precise identification of specific alterations within complex protein landscapes, revealed significant changes in protein expression following TSA treatment under SMG [57]. Notably, we observed a downregulation of key pathways involved in cell proliferation and migration, including proteins associated with G1/S and G2/M cell cycle checkpoints (PA2G4, SFN, YWHAG, YWHAQ) and cytoskeletal dynamics and cell motility (RHOC, RHOA, CDC42, ACTN4, EZR, MYH9, FLNB, FSCN1). Furthermore, proteins involved in cell adhesion (ITGB1, LGALS3, LGALS3BP, SDCBP, PODXL, CDCP1) were also downregulated, suggesting a potential impact on cell–cell and cell–matrix interactions. These findings indicate that TSA under SMG may inhibit tumor growth and metastasis by disrupting these essential cellular processes [58]. We also observed a downregulation of stress response proteins (HSPA1A/HSPA1B, HSPA5, SERPINH1), suggesting that TSA might affect cellular stress response pathways, potentially impacting tumor cell survival. The coordinated downregulation of proteins involved in cytoskeletal dynamics, cell adhesion, and signaling pathways points to a broad disruption of cellular functions critical for tumor progression. The downregulation of tubulin isoforms (TUBA4A, TUBA1B, TUBB, TUBB4B, TUBB6) and actin-related proteins (ACTN4, ARPC5, ACTR2, ACTC1, ACTB), key components of the cytoskeleton, likely impairs cell migration, invasion, and interaction with the microenvironment [59,60]. The downregulation of vinculin (VCL), linking the actin cytoskeleton to integrins, further suggests weakened cell–matrix adhesions, compromising cell motility and invasiveness [61]. Similarly, the downregulation of RAB5C, RAB7A, and DNM2, involved in endocytosis and vesicular trafficking, suggests interference with receptor signaling, nutrient uptake, and protein degradation, essential for tumor growth [62]. The obtained results indicate that TSA treatment disrupts EMT and cytoskeletal organization, both of which are critical for cell migration and invasion. The transcriptional downregulation of Zeb1 and the corresponding upregulation of the E-cadherin protein suggest that TSA promotes a shift from mesenchymal to epithelial characteristics. Furthermore, the proteomic data, showing downregulation of proteins involved in actin-based mobility, including Rho GTPases and actin cytoskeleton components, support the conclusion that TSA impairs PaCa44 cell migration. The observed downregulation of α-tubulin and vinculin, proteins essential for cell structure, adhesion, and migration, reinforces this conclusion. The findings indicate that TSA exerts complex effects on PaCa44 cells, influencing apoptosis, stemness, and cell–cell adhesion. The modulation of multiple apoptotic pathways, coupled with the downregulation of proteins in the PI3k/Akt and HIF-1α pathways, suggests that TSA promotes cell death while inhibiting survival mechanisms. The observed reduction in stemness markers and the disruption of epithelial adherens junctions further implies that TSA may inhibit the cells’ ability to self-renew, migrate, and invade surrounding tissues. Interestingly, while most cytoskeletal proteins were downregulated, α-catenin (CTNNA1), a crucial component of adherens junctions, was upregulated, potentially representing a compensatory mechanism to maintain cell–cell contacts despite overall cytoskeletal disruption [63]. Changes in signaling pathways accompanied these protein expression alterations. The downregulation of Rho GTPase signaling proteins (RhoA, CDC42, and their effectors), which regulate actin dynamics and cell cycle progression [64,65,66,67], aligns with the observed cytoskeletal disruption and likely contributes to the anti-proliferative effects of TSA. The downregulation of cell adhesion proteins (ITGB1, LGALS3, LGALS3BP, SDCBP, PODXL, CDCP1) further suggests a potential impact on tumor cell attachment, migration, and invasion, contributing to the anti-metastatic effects of TSA.

The observed activation of caspase-3, 8, and 9, along with their related mechanistic implications in the current study, are inferred primarily from the analysis of gene expression. While these findings strongly suggest the involvement of specific apoptotic pathways, it is important to note that direct functional validation of these protein activities and their interactions remains a crucial next step. Future studies are planned to experimentally confirm these inferred mechanisms, building upon prior research on caspase expression in other disease contexts [31,68].

In conclusion, our proteomic analysis reveals a complex response to TSA treatment in PDAC cells under SMG, involving significant disruptions in cytoskeletal dynamics, cell adhesion, and signaling pathways. These findings offer valuable insights into the anti-tumor mechanisms of TSA. However, further research is needed to fully elucidate how SMG influences histone acetylation and downstream gene expression, and to explore potential synergistic effects of SMG and TSA with existing chemotherapy regimens to improve PDAC treatment outcomes. Our study highlights the importance of understanding microgravity’s impact on PDAC cells to develop innovative therapeutic approaches against drug resistance and metastasis.

## 4. Materials and Methods

### 4.1. Chemical Agent: Trichostatin A

Trichostatin A (TSA) (T8552 1 mg, lot#088M4163U) was obtained from Sigma-Aldrich Co., Ltd. (Milan, Italy) and dissolved in absolute ethanol at a concentration of 3.3 mM. The stock solution was stored at −80 °C until use.

### 4.2. Cell Culture and Simulated Microgravity Exposure

Human pancreatic ductal adenocarcinoma PaCa44 cells were cultured in RPMI-1640 medium supplemented with 10% fetal bovine serum (FBS) and 1% penicillin–streptomycin (PS). Cells were kindly provided by Prof. M. Donadelli (University of Verona). To simulate microgravity conditions, 5 × 10^5^ cells/mL were seeded in cell culture flask 25 cm^2^ and placed on a Random Positioning Machine (RPM) set to “real random mode”. The RPM continuously rotates at 56 deg/s, minimizing centrifugal acceleration and fluid shear stress. Cells were exposed to SMG for 9 days at 37 °C, and after 7 days of SMG exposure, Trichostatin A was added for a final concentration of 2.5 μM. Experiments were performed on PaCa44 cell spheroids cultured under simulated microgravity conditions. Nine-day simulated microgravity exposure was chosen based on the prior identification of this duration as optimal for observing significant pathway modulations.

### 4.3. Cell Proliferation MTT Assay

Cell viability was assessed using the colorimetric MTT assay (Roche, Indianapolis, IN, USA, 11465007001). Briefly, adherent PaCa44 cells were seeded in 96-well plates at a density of 5 × 10^3^ cells/well in RPMI 1640 medium and allowed to adhere overnight. Subsequently, they were treated for 48 h with increasing concentrations of Trichostatin A (TSA), specifically: 50 nM, 10 nM, 15 nM, 25 nM, 50 nM, 125 nM, 250 nM. Instead, PaCa44 cells spheroids 5 × 10^4^ cells/well were seeded in U-bottom low-adhesion 96-well plates containing RPMI-1640 medium supplemented with 10% methylcellulose. Spheroids were grown at 37 °C and 5% CO_2_ for 9 days, and the last 48 h spheroids were treated with increasing concentrations of TSA, specifically: 0.025 μM, 0.050 μM, 0.125 μM, 0.25 μM, 0.5 μM, 1.0 μM, 2.5 μM, 5.0 μM. Control cells received an equivalent volume of ethanol without TSA. After the desired incubation period, 10 µL of MTT solution (5 mg/mL in PBS) were added to each well and incubated for 4 h at 37 °C. For adherent cells, 100 µL of solubilization solution were used to dissolve the formazan crystals. In contrast, for spheroids, the protocol was slightly modified to improve disaggregation of the spheroid structure: DMSO was used in place of the solubilization solution. The absorbance at 570 nm was measured using a microplate reader. Cell viability was calculated relative to the control group, which was treated with the vehicle alone. The results were calculated as delta absorbance between the samples and the blank condition.

### 4.4. Proteomic Analysis

Cells were collected after 9 days of exposure to simulated microgravity (SMG) and 48 h of treatment with Trichostatin A (TSA). The cells were washed in phosphate-buffered saline (PBS) and resuspended in RIPA buffer (Thermo Fisher Scientific, Waltham, MA, USA) supplemented with a protease inhibitor cocktail (1X, Roche, Basel, Switzerland). To enhance protein yield, the suspension was sonicated twice for 10 min, gently mixed for 15 min on ice, and centrifuged at 14,000× *g* for 15 min at 4 °C to pellet debris. Protein concentration in the supernatant was determined using the BCA Protein Assay (Thermo Fisher Scientific), with bovine serum albumin serving as the standard. Proteins from cell lysates were reduced using 2.5 μL of 200 mM dithiothreitol (DTT) (Sigma-Aldrich, St. Louis, MO, USA) at 90 °C for 20 min, followed by alkylation with 10 μL of 200 mM cysteine-blocking reagent (Iodoacetamide, IAM, Sigma-Aldrich) for 1 h at room temperature in the dark. Digestion was performed overnight at 37 °C using sequence-grade trypsin (Promega, Madison, WI, USA). The resulting peptides were desalted using Discovery^®^ DSC-18 solid-phase extraction (SPE) 96-well plates (25 mg/well, Sigma-Aldrich) and prepared for analysis. The peptide digests were analyzed using a micro-LC Eksigent Technologies system (Dublin, CA, USA) equipped with a Halo Fused C18 column (0.5 × 100 mm, 2.7 μm, Eksigent Technologies). The mobile phase consisted of 0.1% (*v*/*v*) formic acid in water (A) and 0.1% (*v*/*v*) formic acid in acetonitrile (B), with a flow rate of 15.0 μL/min. The gradient elution involved increasing solvent B from 2% to 40% over 30 min. Samples were first analyzed using traditional data-dependent acquisition (DDA) to generate a spectral library. Mass spectrometry was performed over a mass range of 100–1500 Da (TOF scan, 0.25 s accumulation time) with MS/MS product ion scans from 200 to 1250 Da (5.0 ms accumulation time), applying a signal threshold of 30 cps. Subsequently, cyclic data-independent acquisition (DIA) was performed using a 25-Da window. A 50 ms TOF–MS survey scan was followed by MS/MS experiments on all precursors in a cyclic manner (36 swaths, 40 ms accumulation time per swath). Fragmentation was achieved using rolling collision energy. Mass spectrometry data were acquired using Analyst TF v.1.7 (AB SCIEX, Concord, ON, Canada). Three instrumental replicates per sample were subjected to DIA analysis. The data were searched using Protein Pilot software v.4.2 (AB SCIEX) and Mascot v.2.4 (Matrix Science Inc., Boston, MA, USA). The parameters included the UniProt Swiss-Prot human proteins database (version 12/10/2018; 48,561 entries). Search settings: Enzyme: Trypsin (allowing up to 2 missed cleavages); tolerances: ±50 ppm (peptide mass) and ±0.1 Da (MS/MS); modifications: carbamidomethyl cysteine (fixed) and oxidized methionine (variable); peptide charges: +2, +3, and +4. As per validation, there was a false discovery rate (FDR) of 1%. Quantification was performed using the extracted ion chromatograms of unique peptide ions, excluding shared and modified peptides. Results were analyzed with PeakView v.2.0 and MarkerView v.1.2 (AB SCIEX), with six peptides per protein and six transitions per peptide extracted from the SWATH files. Peptides with an FDR < 1.0% were exported to MarkerView for *t*-tests. The processed data were analyzed using Ingenuity Pathways Analysis (IPA, Qiagen, Redwood City, CA, USA) and STRING software (www.stringdb.org—accessed on 13 January 2025). These tools were employed for pathway enrichment and network analysis [58].

### 4.5. Western Blot Analysis

After 9 days of simulated microgravity exposure, PaCa44 cells were collected and lysed with RIPA buffer supplemented with protease and phosphatase inhibitors (PPC1010, Sigma-Aldrich, Milan, Italy). The protein concentration was determined using Bradford protein assay (5000006, Bio-Rad, Segrate, Milano, Italy), and equal amounts of protein were loaded on Mini-PROTEAN TGX stain-free protein gels (4568083, Bio-Rad, Segrate, Milano, Italy) in denaturing and reducing conditions. Proteins were then transferred onto Trans-Blot Turbo mini 0.2 μm nitrocellulose membranes (1704158, Bio-Rad, Segrate, Milano, Italy), then saturated with 5% non-fat milk in PBS- 0.1% Tween 20 buffer, and hybridized overnight at 4 °C with the following primary anti-human antibodies: rabbit anti-E-cadherin (1:1000, Genetex, Irvine, CA, USA), mouse anti-vimentin (1:1000, Santa Cruz, Dallas, TX, USA), mouse anti-GAPDH (1:1000, Santa Cruz, Dallas, TX, USA), mouse anti-αTubulin (1:1000, Merckmillipore, Milano, Italy), mouse anti-β-Actin (1:1000, Cell Signaling, Danvers, MA, USA), mouse anti-vinculin (dilution 1:200; eBioscience, San Diego, CA, USA), mouse anti-Nanog (1:1000, Biorbyt, Cambridge, UK), CD44 (1:1000; Santa-Cruz, Dallas, TX, USA). Membranes were then washed and probed for 1 h at room temperature with HRP-conjugated secondary antibodies quantified by densitometry (ImageLab 6.1 software, Bio-Rad, Segrate, Milano, Italy). Results were normalized by total protein normalization.

### 4.6. RNA Extraction and qRT-PCR

Real-time qPCR quantification was performed through the iTaq Universal SYBR Green Upermix (1725124, Bio-Rad, Segrate, Milano, Italy) with a QuantStudio 3 Real-Time PCR System (Thermo Fisher Scientific, Waltham, MA, USA). The primers used are listed in Table 1. The average of the cycle threshold of each triplicate was analyzed according to the 2^−ΔΔCt^ method. Notably, *18S* gene expression was used as endogenous control to standardize mRNA expression.

### 4.7. Statistical Analysis

Values were expressed as mean ± sem. An unpaired Student’s *t*-test was used to compare unmatched groups with Gaussian distribution. A Mann–Whitney U-test was used in cases of non-Gaussian distribution. *p* ≤ 0.05 was considered statistically significant. Statistical analyses were performed with GraphPad Prism v.8.0.2.

## Figures and Tables

**Figure 1 ijms-26-04758-f001:**
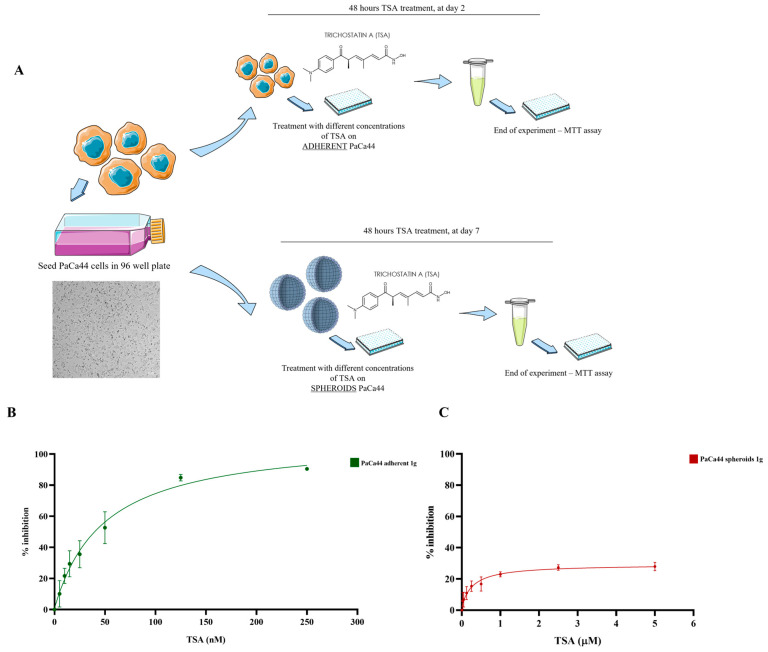
Dose–response curves illustrating the effects of Trichostatin A (TSA) on the viability of PaCa44 cells under two distinct culture conditions. (**A**) represents the experimental set-up; image magnification 115× (**B**) represents cells grown in 2D adherent conditions and treated with TSA for 48 h, a pronounced dose-dependent reduction in cell viability is observed, with a clear inflection point indicating a threshold concentration at which TSA significantly impacts cellular activity; (**C**) shows the response of PaCa44 cells cultured as spheroids (3D culture) always treated with TSA for 48 h. The curve is notably flatter, suggesting a reduced sensitivity to TSA compared to the 2D model. *n* = 4.

**Figure 2 ijms-26-04758-f002:**
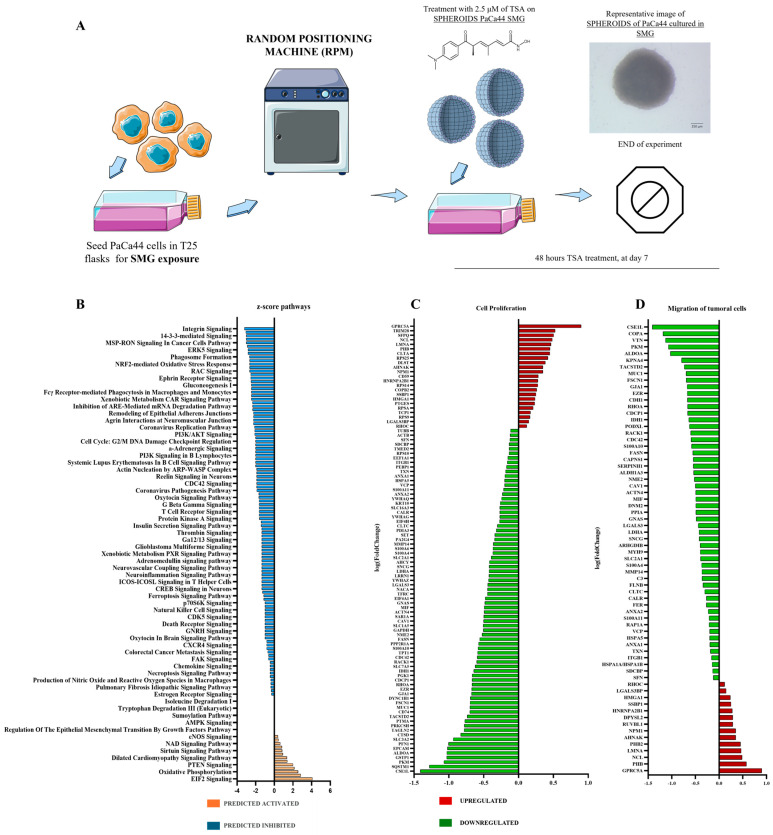
Exposure of PaCa44 cells to simulated microgravity (SMG): (**A**) experimental workflow: PaCa44 cells were subjected to simulated microgravity (SMG) for a total duration of 9 days, with Trichostatin A (TSA) administered at a concentration of 2.5 μM during the final 48 h of the experiment; (**B**) canonical pathways modulated by differentially expressed proteins in PaCa44 cells after 48 h of TSA treatment and 9 days SMG exposure; (**C**) cell proliferation signaling evaluated in PaCa44 cells maintained in SMG for 9 days and 48 h of TSA treatment; (**D**) signaling of migration of tumoral cells in PaCa44 cells maintained in SMG for 9 days and 48 h of TSA treatment.

**Figure 3 ijms-26-04758-f003:**
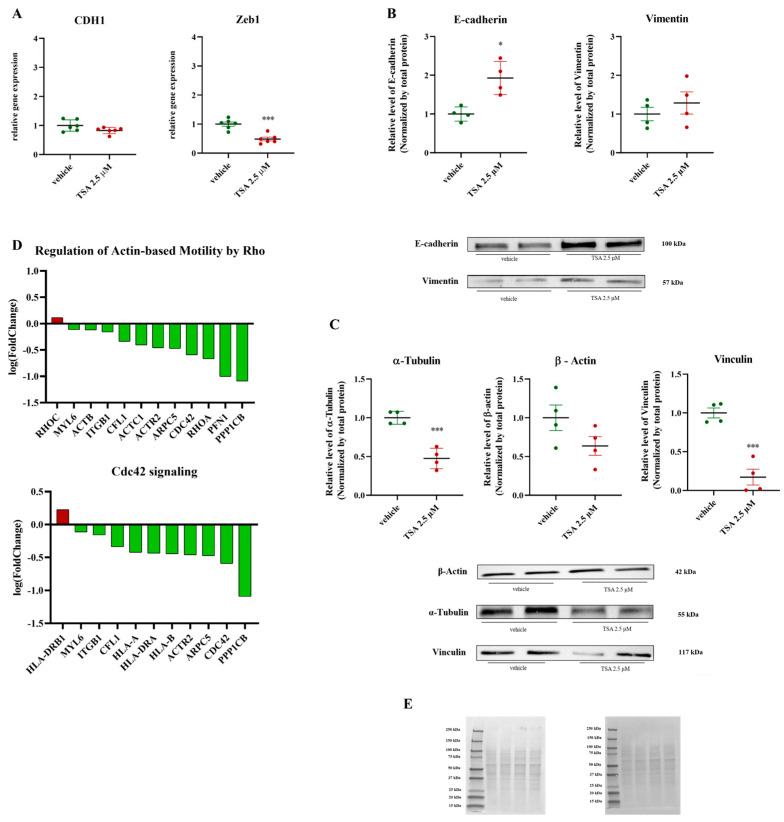
Alterations in cell adhesion and migration pathways in PaCa44 cell line exposed to simulated microgravity for 9 days, with a 48 h treatment with 2.5 µM Trichostatin A. (**A**): RT-qPCR analysis of E-cadherin (CDH1) and Zeb1, EMT-associated markers in PaCa44 cells treated with 2.5 µM of TSA, Data are expressed as mean ± sem of *n* = six replicates. *** *p* < 0.001; (**B**,**C**): Western blot analysis of E-cadherin (CDH1), Vimentin, cytoskeletal proteins, including α-tubulin, β-actin, and vinculin in PaCa44 cells treated with 2.5 µM of TSA. Results were normalized by total protein normalization of Ponceau S, Data are expressed as mean ± sem of *n* = four replicates. * *p* < 0.05; *** *p* < 0.001; (**D**): Average logarithmic fold change in proteins involved in actin-based motility (Rho signaling) and Cdc42 signaling pathways. In green proteins compared to the control group (9 days of SMG); (**E**): Ponceau S-stained membrane illustrating total protein for normalization.

**Figure 4 ijms-26-04758-f004:**
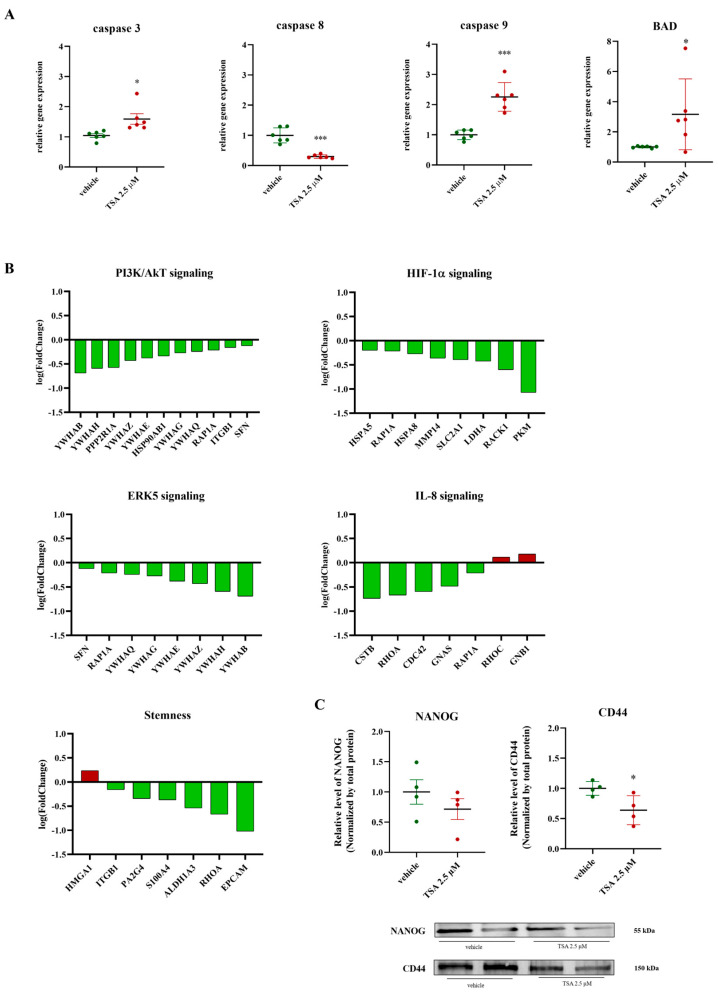
TSA (2.5 μM) treatment modulates apoptosis, PI3K/AKT, ERK, IL-8, and HIF-1α signaling pathways and affects stemness markers. (**A**) Relative gene expression levels of caspase 3, caspase 8, caspase 9 and BAD; in PaCa44 cells treated with 2.5 µM of TSA, data are expressed as mean ± sem of *n* = 6 replicates. * *p* < 0.05; *** *p* < 0.0001; (**B**) average logarithmic fold change in proteins involved in PI3K/AKT signaling, ERK signaling, IL-8 signaling, and HIF-1α signaling; (**C**) Western blot analysis of NANOG and CD44 in PaCa44 cells treated with 2.5 µM of TSA. Results were normalized by total protein normalization of Ponceau S, Data are expressed as mean ± sem of *n* = four replicates. * *p* < 0.05. Ponceau S-stained membrane illustrating total protein for normalization in Figure 3E.

**Figure 5 ijms-26-04758-f005:**
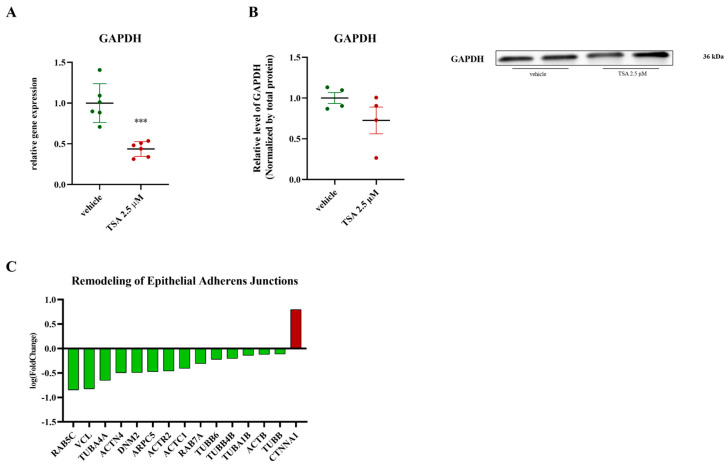
Effects of TSA on GAPDH and epithelial adherens junction genes. (**A**) Relative gene expression levels GPDH in PaCa44 cells treated with 2.5 µM of TSA, data are expressed as mean ± sem of *n* = 6 replicates. *** *p* < 0.001; (**B**) Western blot analysis of GAPDH in PaCa44 cells treated with 2.5 µM of TSA. Results were normalized by total protein normalization of Ponceau S; (**C**): average logarithmic fold change in proteins involved in remodeling of epithelial adherens junctions signaling. Ponceau S-stained membrane illustrating total protein for normalization in Figure 3E.

**Table 1 ijms-26-04758-t001:** Sequences of primers used for qPCR analysis.

Genes	Forward Primer Sequence	Reverse Primer Sequence
*CDH1*	5′-GAC ACC AAC GAT AAT CCT CCGA-3′	5′-GGC ACC TGA CCC TTG TAC GT-3′
*ZEB1*	5′-GTT ACC AGG GAG GAG CAG TGAAA-3′	5′-GAC AGC AGT GTC TTG TTG TTG TAG AAA-3′
*caspase 3*	5′-CTGGTTTTCGGTGGGTGT-3′	5′-CACTGAGTTTTCAGTGTTCTCCA-3′
*caspase 8*	5′-CAGCAGCCTTGAAGGAAGTC-3′	5′-CGAGATTGTCATTACCCCACA-3
*caspase 9*	5′-CCCAAGCTCTTTTTCATCCA-3′	5′-AGTGGAGGCCACCTCAAAC-3′
*BAD*	5′-ACCAGCAGCAGCCATCAT-3′	5′-GGTAGGAGCTGTGGCGACT-3′
*GAPDH*	5′-ATC AGC AAT GCC TCC TGC AC-3′	5′-TGG TCA TGA GTC CTT CCA CG-3′
*18S*	5′-ACT TTC GAT GGT AGT CGC CGT-3′	5′-CCT TGG ATG TGG TAG CCG TTT-3′

## Data Availability

Raw data are available in Mendeley Data Repository as https://data.mendeley.com/datasets/cn22f6pkzv/1 (created 9 April 2025) (DOI: 10.17632/cn22f6pkzv.1).

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
