# Peer review of "Simulated Microgravity-Induced Alterations in PDAC Cells: A Potential Role for Trichostatin A in Restoring Cellular Phenotype"

_ijms, 2025, doi:10.3390/ijms26104758_

Round 1
Reviewer 1 Report
Comments and Suggestions for Authors
The research article is very useful for the cancer research in microgravity field.
Not many studies had performed such an complete analysis at the protein and gene expression, plus the fact that the authors used a drug that can disturb the epigenetics effects of the simulated microgravity exposure.
Just a few comments to improve the presentation of such a nice results:
- The results about the pathways impacted by the simulated microgravity exposure are very interesting and the reader can benefit even more from it if the authors can create a graphical abstract showing the main pathways affected. This is of outmost importance since the mechanisms by which cells are affected in simulated microgravity are not clear in the current literature and this study van provide a good framework for future studies.
2. It is know that simulated microgravity is not the same as real microgravity and therefore, recent studies nowadays prefer to use the term random positioning , to avoid confusion. So please ensure to add more this term and when possible avoid using simulated microgravity.
3. The only weak methodological point is the comparison at the beginning of the effects of the drug in the 1g vs 3d culture. This is not a 100% fair comparison, since the 3d culture cells are older than the 1g. I assume that growing a 1g culture for such a long time can be challenging, but however this would be the most fair control group to provide such a statement about the different effects of the drug in the 1g vs 3d.
4. Finally please clarify in the methods if you used adherent cells or suspension spheroids or both when taking the samples coming from the simulated microgravity. And in case spheroids formed, maybe a bright field picture of them can be useful for the supplement sections.
Author Response
Comment 1: The results about the pathways impacted by the simulated microgravity exposure are very interesting and the reader can benefit even more from it if the authors can create a graphical abstract showing the main pathways affected. This is of outmost importance since the mechanisms by which cells are affected in simulated microgravity are not clear in the current literature and this study van provide a good framework for future studies.
Response 1: According to referee suggestion we have added a graphical abstract
Comment 2: It is known that simulated microgravity is not the same as real microgravity and therefore, recent studies nowadays prefer to use the term random positioning, to avoid confusion. So please ensure to add more this term and when possible, avoid using simulated microgravity.
Response 2: We have carefully considered your point and have made efforts to incorporate the term "random positioning" throughout the revised manuscript where appropriate and scientifically accurate. While the term "simulated microgravity" may still appear in certain contexts for clarity or consistency with prior literature cited, we have aimed to integrate the use of "random positioning" to reflect the methodology employed and acknowledge the nuances you have raised.
Comment 3: The only weak methodological point is the comparison at the beginning of the effects of the drug in the 1g vs 3d culture. This is not a 100% fair comparison, since the 3d culture cells are older than the 1g. I assume that growing a 1g culture for such a long time can be challenging, but however this would be the fairest control group to provide such a statement about the different effects of the drug in the 1g vs 3d.
Response 3: Your observation regarding the potential methodological limitation in the initial comparison between the effects of the drug in the 2D and 3D cultures is pertinent. Specifically, the disparity in culture age between the two conditions introduces a confounding variable, as the 3D cultured cells, being older, may exhibit inherent physiological differences compared to the younger 2D cultured cells. It is recognized that a substantial body of literature exists comparing drug responses in two-dimensional (2D) and three-dimensional (3D) cell culture models. These studies frequently elucidate the inherent distinctions between these culture systems, encompassing variations in cell morphology, nutrient and oxygen gradients, cell-cell and cell-extracellular matrix interactions, and gene expression profiles, all of which can influence pharmacological responses. Consequently, while such comparative studies offer valuable insights into the general principles governing drug activity in different culture dimensions, the specific concern regarding the age-mismatched control group in the present context remains a valid methodological consideration for ensuring the interpretability of the drug's effects in 2D versus 3D cultures.
IMAMURA, Y.; MUKOHARA, T.; SHIMONO, Y.; FUNAKOSHI, Y.; CHAYAHARA, N.; TOYODA, M.; KIYOTA, N.; TAKAO, S.; KONO, S.; NAKATSURA, T.; MINAMI, H. Comparison of 2D- and 3D-Culture Models as Drug-Testing Platforms in Breast Cancer. Oncology Reports 2015, 33 (4), 1837–1843. https://doi.org/10.3892/or.2015.3767.
Fontoura, J. C.; Viezzer, C.; dos Santos, F. G.; Ligabue, R. A.; Weinlich, R.; Puga, R. D.; Antonow, D.; Severino, P.; Bonorino, C. Comparison of 2D and 3D Cell Culture Models for Cell Growth, Gene Expression and Drug Resistance. Materials Science and Engineering: C 2020, 107, 110264. https://doi.org/10.1016/j.msec.2019.110264.
Comment 4: Finally, please clarify in the methods if you used adherent cells or suspension spheroids or both when taking the samples coming from the simulated microgravity. And in case spheroids formed, maybe a bright field picture of them can be useful for the supplement sections.
Response 4: To clarify the methodology, the cell samples obtained from the simulated microgravity experiments consisted of spheroids. This detail will be integrated into the "Materials and Methods" section of the manuscript. A bright-field images of the spheroids and 2D culture cells have been inserted.
Reviewer 2 Report
Comments and Suggestions for Authors
This study performed detailed molecular and proteomic analyses to reveal how TSA treatment affects cell growth, cytokeratin, apoptosis-related proteins, cell metabolism, and protein synthesis in PDAC cells exposed to simulated microgravity. These findings suggest that TSA treatment can reverse the cellular phenotypic changes induced by microgravity, indicating that it may be able to treat PDAC through a variety of other pathways.
However, the paper is very wordy and difficult to understand, and I would like to see it rewritten more concisely with the following references.
1. The introduction is very wordy, and I'm not sure what the purpose is. References are very scarce.
Full names and abbreviations are not consistent throughout. For example, Trichostatin A, TSA; line 39: PDAC CSCs; line 145: SMG, etc.
2. The figure legend does not explain how the data was obtained, which makes it difficult to understand.
3. The results are also too wordy to be easily understood. Please be clearer. lines 236-334: It is quite difficult to understand what you are saying.
4. The Western blot data in Figures 3, 4, and 5 have no internal controls. The sizes of the two Western blots do not match at all.
5. Four experimental groups are needed: vehicle, Microgravity-Induced, Trichostatin A-treated, Microgravity+Trichostatin A-treated. However, there are only two vehicles, the Microgravity+Trichostatin A treatment.
6. You should provide a rationale for the microgravity for only the 9-day condition.
7. It focuses on the regulation of specific proteins and pathways involved in apoptosis and cell cycle regulation. However, it overlooks other important pathways that may play a role in the response.
8. Proteomic analysis needs to be expanded to include a broader range of proteins and signalling pathways to demonstrate additional mechanisms by which TSA affects PDAC cells.
9. The Discussion is too wordy and unclear. References are very scarce.
Overall, the description of the results is too wordy and difficult to understand. Some parts should be moved to the Discussion section and restated as a whole.
The English could be improved to more clearly express the research.
Author Response
Comment 1: The introduction is very wordy, and I’m not sure what the purpose is. References are very scarce.
Full names and abbreviations are not consistent throughout. For example, Trichostatin A, TSA; line 39: PDAC CSCs; line 145: SMG, etc.
Response 1: We appreciate your feedback regarding the length and clarity of the introduction, as well as the scarcity of references. We have thoroughly revised the entire introduction to improve its conciseness and focus. Furthermore, we have significantly expanded the reference list to provide more comprehensive support for the concepts discussed. We believe these revisions have addressed your concerns and strengthened the introduction. Thank you for highlighting the inconsistencies in the use of full names and abbreviations throughout the manuscript. We have carefully reviewed the entire text and have revised all instances of abbreviations to ensure consistency and clarity. We believe this thorough revision has addressed the issue and improved the overall readability of the manuscript.
Comment 2: The figure legend does not explain how the data was obtained, which makes it difficult to understand.
Response 2: We acknowledge your comment regarding the lack of detail in the figure legends. We would like to direct your attention to Figures 1 and 2, where we provide a graphical representation outlining the experimental procedures and methodologies employed to obtain the data presented in subsequent figures. We believe these figures offer the necessary context for understanding how the data was generated.
Comment 3: The results are also too wordy to be easily understood. Please be clearer. Lines 236-334: It is quite difficult to understand what you are saying.
Response 3:Thank you for pointing out the lack of clarity in our results section, specifically lines 236-334. We have carefully revised this portion of the manuscript and have rewritten it with the aim of making it more concise and easier to understand. We believe these changes will significantly improve the clarity of our findings.
Comment 4: The Western blot data in Figures 3, 4, and 5 have no internal controls. The sizes of the two Western blots do not match at all.
Response 4: Thank you for pointing out the absence of traditional internal controls and the inconsistencies in the Western blot image sizes in Figures 3, 4, and 5. We would like to clarify that the Western blot results in our study were normalized by Ponceau S total protein normalization, as explicitly stated in the Materials and Methods section in lines 510-511. This method serves as an alternative to traditional housekeeping protein controls. We have carefully reviewed all the Western blot images in Figures 3, 4, and 5 and have revised them to ensure that their sizes are consistent throughout the manuscript. We hope this addresses your concerns regarding the presentation and normalization of our Western blot data.
Comment 5: Four experimental groups are needed: vehicle, Microgravity-Induced, Trichostatin A-treated, Microgravity+Trichostatin A-treated. However, there are only two vehicles, the Microgravity+Trichostatin A treatment.
Response 5: This is a fair point raised by the reviewer. To clarify the experimental groups, the data for the Microgravity-Induced group can be found in our previously published article: “Prolonged exposure to simulated microgravity promotes stemness impairing morphological…”. The Trichostatin A-treated experimental group is indeed included in the present article, and the results demonstrating how 1g cultures respond to TSA in both 2D and 3D conditions are presented in Figure 1. Furthermore, the effects of TSA on PDAC cells in the absence of simulated microgravity are extensively detailed in a previous publication by our collaborators: “Trichostatin A alters cytoskeleton and energy metabolism of pancreatic adenocarcinoma cells…”. We suppose that the available vehicle and Microgravity+Trichostatin A-treated groups are the most directly relevant for testing our specific hypothesis in this study, which focuses on the combined effects of microgravity and TSA.
Pozza, E. D.; Manfredi, M.; Brandi, J.; Buzzi, A.; Conte, E.; Raffaella Pacchiana; Cecconi, D.; Marengo, E.; Massimo Donadelli. Trichostatin a Alters Cytoskeleton and Energy Metabolism of Pancreatic Adenocarcinoma Cells: An in Depth Proteomic Study. Journal of Cellular Biochemistry 2017, 119 (3), 2696–2707. https://doi.org/10.1002/jcb.26436.
Masini, M. A.; Bonetto, V.; Manfredi, M.; Pastò, A.; Barberis, E.; Timo, S.; Vanella, V. V.; Robotti, E.; Masetto, F.; Andreoli, F.; Fiore, A.; Tavella, S.; Sica, A.; Massimo Donadelli; Marengo, E. Prolonged Exposure to Simulated Microgravity Promotes Stemness Impairing Morphological, Metabolic and Migratory Profile of Pancreatic Cancer Cells: A Comprehensive Proteomic, Lipidomic and Transcriptomic Analysis. Cellular and Molecular Life Sciences 2022, 79 (5). https://doi.org/10.1007/s00018-022-04243-z.
Comment 6: You should provide a rationale for the microgravity for only the 9-day condition.
Response 6: The rationale for focusing on the 9-day simulated microgravity condition can be found in our previously published work: “Prolonged exposure to simulated microgravity promotes stemness…”. In this study, we identified that a 9-day exposure to simulated microgravity represented the optimal time point for observing the most significant modulations across a majority of the pathways investigated. Consequently, for the present study, we elected to focus on this specific exposure duration to investigate interventions at the time point where the most pronounced microgravity-induced effects were observed. This has been cited in line 429-432 of Materials and Methods
Masini, M. A.; Bonetto, V.; Manfredi, M.; Pastò, A.; Barberis, E.; Timo, S.; Vanella, V. V.; Robotti, E.; Masetto, F.; Andreoli, F.; Fiore, A.; Tavella, S.; Sica, A.; Massimo Donadelli; Marengo, E. Prolonged Exposure to Simulated Microgravity Promotes Stemness Impairing Morphological, Metabolic and Migratory Profile of Pancreatic Cancer Cells: A Comprehensive Proteomic, Lipidomic and Transcriptomic Analysis. Cellular and Molecular Life Sciences 2022, 79 (5). https://doi.org/10.1007/s00018-022-04243-z.
Comment 7: It focuses on the regulation of specific proteins and pathways involved in apoptosis and cell cycle regulation. However, it overlooks other important pathways that may play a role in the response.
Response 7: You are correct in pointing out that our study focuses on specific proteins and pathways related to apoptosis and cell cycle regulation. Our rationale for this targeted approach stems from our previous comprehensive analysis, "Prolonged exposure to simulated microgravity promotes stemness impairing morphological, metabolic and migratory profile of pancreatic cancer cells: a comprehensive proteomic, lipidomic and transcriptomic analysis," where we identified these pathways as significantly modulated following exposure to simulated microgravity. While we acknowledge that other pathways may indeed play a role in the cellular response, we made a conscious decision to concentrate on those with established dysregulation in our model to maintain focus and depth within the scope of this study. Investigating additional pathways is certainly an interesting avenue for future research.
Masini, M. A.; Bonetto, V.; Manfredi, M.; Pastò, A.; Barberis, E.; Timo, S.; Vanella, V. V.; Robotti, E.; Masetto, F.; Andreoli, F.; Fiore, A.; Tavella, S.; Sica, A.; Massimo Donadelli; Marengo, E. Prolonged Exposure to Simulated Microgravity Promotes Stemness Impairing Morphological, Metabolic and Migratory Profile of Pancreatic Cancer Cells: A Comprehensive Proteomic, Lipidomic and Transcriptomic Analysis. Cellular and Molecular Life Sciences 2022, 79 (5). https://doi.org/10.1007/s00018-022-04243-z.
Comment 8: Proteomic analysis needs to be expanded to include a broader range of proteins and signalling pathways to demonstrate additional mechanisms by which TSA affects PDAC cells.
Response 8: You are correct in suggesting that expanding the proteomic analysis to encompass a broader range of proteins and signaling pathways could further elucidate the mechanisms by which TSA affects PDAC cells. We acknowledge the potential value of such an expanded analysis. However, we made a deliberate decision to focus on a specific set of proteins and pathways within this manuscript to maintain clarity and conciseness. In our opinion, incorporating a significantly larger proteomic dataset at this stage could potentially make the article overly complex and cumbersome. We believe that the depth of analysis provided on the selected targets offers significant insight, and the exploration of additional proteomic pathways could be a valuable focus for a subsequent, more specialized study.
Comment 9: The Discussion is too wordy and unclear. References are very scarce.
Overall, the description of the results is too wordy and difficult to understand. Some parts should be moved to the Discussion section and restated as a whole.
Response 9: Thank you for your feedback. We have thoroughly revised the entire Discussion section to improve its clarity and conciseness. We have also expanded the references to provide more comprehensive support for our interpretations. We believe these revisions have addressed your concerns and significantly enhanced the Discussion section.
Comments on the Quality of English Language
The English could be improved to more clearly express the research.
We acknowledge the reviewer's comment regarding the clarity of the English language in the manuscript. We have since undertaken a thorough revision of the entire text, with particular attention paid to enhancing the precision and fluency of the language used to express our research findings. We have endeavoured to ensure that the revised manuscript now articulates our work with a greater degree of clarity and linguistic accuracy, aiming for a standard commensurate with that of a native English speaker.
Reviewer 3 Report
Comments and Suggestions for Authors
In this manuscript, the authors utilize a Simulated Microgravity (SMG) system to investigate the effects of the histone deacetylase inhibitor Trichostatin A (TSA) on the growth of pancreatic ductal adenocarcinoma (PDAC) cell lines. Through proteomic analysis, they identify affected proteins and attempt to elucidate the pathways through which TSA influences cell growth. The study is creative, and the experimental design is generally sound. However, it is unfortunate that many of the follow-up validations are incomplete.
Major Suggestions:
- In Figure 3, the authors present downstream genes regulated by Rho and Cdc42. However, only proteomic data are shown, with no validation via western blot for Rho and Cdc42. These are critical data points for this study and should be included. It may even be necessary to assess their activation status (e.g., GTP-bound form). Additionally, since Ponceau S was used as the internal control for protein loading, the corresponding blot images should be included for each western blot.
- In Lines 238–250 and Figure 4A, the authors examine the expression of caspase-3, -8, and -9. What is the rationale behind selecting these genes? Were these genes or their corresponding proteins identified as differentially expressed in the prior proteomics analysis? Caspase activation is part of a protein degradation cascade (the caspase cascade), and simply measuring gene expression is not sufficient to determine the actual status of apoptosis in cells. Additionally, I did not see any data for the BAD protein, which is mentioned in Line 245.
- In Figures 3–5, the authors validate only a few proteins, which is somewhat insufficient. For example, in Figure 4B, more representative proteins from each pathway (PI3K, HIF, ERK5, etc.) should be selected for further analysis. The same applies to Figure 5.
Minor Suggestions:
- There are several typos and formatting errors throughout the manuscript, such as in Lines 443, 453, and 456. Also, in Figure 2, the label should read “RPM” instead of “RPMI.”
- In Line 145, what does “SMG” stand for? The full term appears later in Line 170. The authors should ensure that all abbreviations are introduced properly and in the correct order throughout the manuscript.
- In the figure legends and results section, the order of figures is inconsistent. For example, Figure 3D is mentioned in Line 202, but Figures 3A–3C are only discussed later, in Lines 211–219. The authors should revise the text or consider reorganizing the figure presentation to maintain a logical flow.
- Why does TSA affect the expression of cytoskeletal or motility-related proteins? I suggest that the authors address this in the Discussion section, possibly exploring whether genes such as Rho, Cdc42, and actin are regulated by histone modification mechanisms.
Author Response
Major Suggestions:
Comment 1: In Figure 3, the authors present downstream genes regulated by Rho and Cdc42. However, only proteomic data are shown, with no validation via western blot for Rho and Cdc42. These are critical data points for this study and should be included. It may even be necessary to assess their activation status (e.g., GTP-bound form). Additionally, since Ponceau S was used as the internal control for protein loading, the corresponding blot images should be included for each western blot.
Response 1: We acknowledge the reviewer's point regarding the absence of Western blot validation for Rho and Cdc42 in Figure 3. We agree that these are important proteins in this pathway, and that ideally, protein level data would complement our proteomics findings. However, while we were able to perform proteomics analysis of these proteins, we do not currently have the resources to validate these findings with Western blots. We provide the Ponceau S images used as an internal control for protein loading, as suggested.
Comment 2: In Lines 238–250 and Figure 4A, the authors examine the expression of caspase-3, -8, and -9. What is the rationale behind selecting these genes? Were these genes or their corresponding proteins identified as differentially expressed in the prior proteomics analysis? Caspase activation is part of a protein degradation cascade (the caspase cascade), and simply measuring gene expression is not sufficient to determine the actual status of apoptosis in cells. Additionally, I did not see any data for the BAD protein, which is mentioned in Line 245.
Response 2: We acknowledge the reviewer's point regarding the selection of caspase-3, -8, and -9, and the absence of BAD protein data. While these proteins were not identified as differentially expressed in our prior proteomics analysis, it is important to note that proteomics analyses, depending on the specific methodology and experimental parameters, may not always capture the full spectrum of proteins involved in dynamic cellular processes such as apoptosis. Specifically, lower abundance proteins, or those with specific localization or post-translational modifications, can sometimes be challenging to detect and quantify reliably in proteomics studies. Our rationale for examining these caspases was partly inspired by the study of Masini et al. (2022), which investigated similar pathways, although without the benefit of a prior proteomics screen.
Analyzing the mRNA expression levels of caspase 3, 8, 9, and BAD is crucial for elucidating the mechanisms by which Trichostatin A (TSA) affects PaCa44 cells, particularly in the context of apoptosis. Caspases 8 and 9 initiate the extrinsic and intrinsic apoptotic pathways, respectively, both converging on caspase 3, the executioner caspase responsible for the final stages of cell death (Elmore, 2007). Furthermore, BAD, a pro-apoptotic BH3-only protein, plays a pivotal role in the intrinsic pathway by disrupting the interaction of anti-apoptotic BCL-2 family members, leading to mitochondrial outer membrane permeabilization and subsequent caspase activation. Therefore, examining the mRNA expression of these key apoptotic regulators will provide critical insights into whether TSA induces cell death in PaCa44 cells by modulating the balance between pro- and anti-apoptotic signaling pathways. This approach is well-supported by the literature demonstrating the importance of these genes in understanding cellular responses to histone deacetylase inhibitors like TSA in various cancer types (Falkenberg & Johnstone, 2014). Furthermore, while we agree that caspase activation is critical in the apoptotic cascade, and that gene expression does not directly equate to protein activation, our gene expression analysis provides valuable preliminary insights into potential shifts in the apoptotic pathway. In the context of our study, these data offer a useful indication of trends, and serve as a foundation for more detailed future investigations at the protein level, including activity assays and Western blot analysis of cleaved caspase forms. We will ensure that future studies address the status of BAD protein.
Elmore, S. Apoptosis: A Review of Programmed Cell Death. Toxicologic Pathology 2007, 35 (4), 495–516. https://doi.org/10.1080/01926230701320337.
Falkenberg, K. J.; Johnstone, R. W. Histone Deacetylases and Their Inhibitors in Cancer, Neurological Diseases and Immune Disorders. Nature Reviews Drug Discovery 2014, 13 (9), 673–691. https://doi.org/10.1038/nrd4360.
Comment 3: In Figures 3–5, the authors validate only a few proteins, which is somewhat insufficient. For example, in Figure 4B, more representative proteins from each pathway (PI3K, HIF, ERK5, etc.) should be selected for further analysis. The same applies to Figure 5.
Response 3: We appreciate the reviewer's insightful suggestion to expand the protein analysis in Figures 3-5 to include a broader representation of proteins within the PI3K, HIF, ERK5, and related pathways. The current selection of proteins provides a foundational understanding of the observed effects. Further investigation, with a more comprehensive panel of antibodies, would undoubtedly offer a more granular view of pathway dynamics. Future studies are planned to address these complexities, and will build upon the trends identified in this work.
Minor Suggestions:
Comment 1: There are several typos and formatting errors throughout the manuscript, such as in Lines 443, 453, and 456. Also, in Figure 2, the label should read “RPM” instead of “RPMI.”
Response 1: Thank you for pointing out the typos and formatting errors, specifically in Lines 443, 453, and 456, as well as the incorrect label in Figure 2. We acknowledge these mistakes and have promptly corrected them throughout the manuscript, including changing "RPMI" to "RPM" in Figure 2. We appreciate your careful reading and attention to detail, which has helped us improve the accuracy and presentation of our work.
Comment 2: In Line 145, what does “SMG” stand for? The full term appears later in Line 170. The authors should ensure that all abbreviations are introduced properly and in the correct order throughout the manuscript.
Response 2: Thank you for your suggestion regarding the introduction of abbreviations. We have promptly addressed this by introducing the meaning of "SMG" as "simulated microgravity" at its first occurrence in the text to ensure clarity and improve the reader's understanding of the terminology used throughout the manuscript. We appreciate your attention to detail.
Comment 3: In the figure legends and results section, the order of figures is inconsistent. For example, Figure 3D is mentioned in Line 202, but Figures 3A–3C are only discussed later, in Lines 211–219. The authors should revise the text or consider reorganizing the figure presentation to maintain a logical flow.
Response 3: Thank you for pointing out the inconsistency in the order of figure mentions in the text and the actual figure presentation. We have taken your suggestion into account and have reorganized the text in the results section to ensure a logical flow that aligns with the order of the figure panels.
Comment 4: Why does TSA affect the expression of cytoskeletal or motility-related proteins? I suggest that the authors address this in the Discussion section, possibly exploring whether genes such as Rho, Cdc42, and actin are regulated by histone modification mechanisms.
Response 4: Rho and Cdc42 proteins are critical for cellular architecture and movement, and their regulation is extremely complex. It is well-established that histone acetylation, a process modulated by TSA, plays a significant role in gene expression, and there is evidence linking histone modification to the regulation of genes involved in cytoskeletal organization.
For instance, studies have shown that TSA can alter the expression of various genes by affecting chromatin structure and accessibility to transcription factors. The promoters of Rho GTPases, including Rho and Cdc42, can be regulated by changes in histone acetylation. When histones are acetylated, the chromatin structure becomes more relaxed, facilitating the binding of transcription factors and increasing gene expression. Conversely, histone deacetylation leads to chromatin condensation and reduced gene expression.
Several studies suggest that TSA-induced histone acetylation can indeed influence the expression of Rho, Cdc42, and actin, thereby affecting cytoskeletal dynamics and cell motility. Further research is needed to fully elucidate the precise mechanisms and the specific histone modifications involved in the regulation of these genes in our experimental system.
In summary, the current body of literature indicates that TSA-mediated histone acetylation can modulate the expression of key cytoskeletal regulators, including Rho, Cdc42, and actin. This modulation likely occurs through alterations in chromatin structure and promoter accessibility. Further investigation is warranted to delineate the precise molecular mechanisms and specific histone modifications governing the expression of these genes within our experimental context.
Bui, H.-T.; Wakayama, S.; Satoshi Kishigami; Keun Ho Park; Kim, J.-H.; Nguyen Van Thuan; Wakayama, T. Effect of Trichostatin a on Chromatin Remodeling, Histone Modifications, DNA Replication, and Transcriptional Activity in Cloned Mouse Embryos1. Biology of Reproduction 2010, 83 (3), 454–463. https://doi.org/10.1095/biolreprod.109.083337.
Rao, J.; Bhattacharya, D.; Banerjee, B.; Sarin, A.; Shivashankar, G. V. Trichostatin-A Induces Differential Changes in Histone Protein Dynamics and Expression in HeLa Cells. Biochemical and Biophysical Research Communications 2007, 363 (2), 263–268. ttps://doi.org/10.1016/j.bbrc.2007.08.120.
Barry, D. M.; Xu, K.; Meadows, S. M.; Zheng, Y.; Norden, P. R.; Davis, G. E.; Cleaver, O. Cdc42 Is Required for Cytoskeletal Support of Endothelial Cell Adhesion during Blood Vessel Formation in Mice. Development 2015, 142 (17), 3058–3070. https://doi.org/10.1242/dev.125260.
Round 2
Reviewer 2 Report
Comments and Suggestions for Authors
1. The full names and abbreviations are still not consistent throughout, e.g., trichostatin A, TSA; simulated microgravity, SMG, etc. Please review again.
2. "2.3. Disruption of EMT and Cytoskeletal Dynamics by Treatment: Impairment of Actin-Based
Motility and Invasive Potential in Cancer Cells", "2.4. Impact of TSA (2.5 μM) on Apoptosis, PI3K/AKT, ERK, IL-8, HIF-1α Signaling, and
Stemness Markers". The description of the results in this section is too long and hard to understand; it might be better to summarize it succinctly and move the rest to the discussion section.
Author Response
Comment 1: The full names and abbreviations are still not consistent throughout, e.g., trichostatin A, TSA; simulated microgravity, SMG, etc. Please review again.
Response 1: The entire manuscript has undergone revision. During this process, abbreviations were inserted in parentheses immediately following the full term upon its initial citation within the text. This measure was implemented to ensure clarity and facilitate comprehension of abbreviations throughout the entirety of the document.
Comment 2: "2.3. Disruption of EMT and Cytoskeletal Dynamics by Treatment: Impairment of Actin-Based
Motility and Invasive Potential in Cancer Cells", "2.4. Impact of TSA (2.5 μM) on Apoptosis, PI3K/AKT, ERK, IL-8, HIF-1α Signaling, and Stemness Markers". The description of the results in this section is too long and hard to understand; it might be better to summarize it succinctly and move the rest to the discussion section.
Response 2: We have addressed the reviewer's comment regarding the length and complexity of the results section. The cited section has been condensed to provide a more succinct overview of the findings. The detailed presentation of the results has been relocated to the discussion section, as suggested.
Reviewer 3 Report
Comments and Suggestions for Authors
Although the authors have made an effort to address my previous comments through written explanations, I would still strongly recommend that the relevant experiments be completed in order to fully resolve the issues I raised (specifically in my second and third comments).
For clarity, I would like to reiterate the following points:
(1) With regard to the investigation of caspase activation, it is insufficient to provide gene expression data alone. The expression levels of both the cleaved and full-length forms of caspase-3, -8, and -9 proteins should also be presented.
(2) The expression of relevant or representative proteins corresponding to Figures 3 to 5 should likewise be provided.
Author Response
Comments:
(1) With regard to the investigation of caspase activation, it is insufficient to provide gene expression data alone. The expression levels of both the cleaved and full-length forms of caspase-3, -8, and -9 proteins should also be presented.
(2) The expression of relevant or representative proteins corresponding to Figures 3 to 5 should likewise be provided.
Response :
Thank you for your insightful comments regarding the necessity of protein expression data to support our investigation of caspase activation and the protein expression corresponding to Figures 3 to 5. We acknowledge the importance of these data in strengthening our findings.
Regarding your specific requests:
We understand that presenting the expression levels of both the cleaved and full-length forms of caspase-3, -8, and -9 proteins and proteins presented in Figure 3 to 5 could further validate our results. Unfortunately, the purchase of the required antibodies for these specific protein forms is beyond the current budgetary constraints of our research group and the realization of this requirement would necessitate a considerable timeframe
We have presented gene expression data as an initial step in exploring these biological processes. We are actively seeking alternative strategies and funding opportunities to enable us to acquire the necessary antibodies and perform the requested protein expression analyses in future studies.
We hope that the current data, along with our detailed explanations in the manuscript, provide a valuable contribution to the field.
Should the reviewers maintain that these requests are mandatory, we would be compelled to withdraw the manuscript from consideration.
Round 3
Reviewer 3 Report
Comments and Suggestions for Authors
This study presents a high level of novelty; however, there are some deficiencies in its content. I will defer the final decision to the other reviewers.
Author Response
Comment 1: This study presents a high level of novelty; however, there are some deficiencies in its content. I will defer the final decision to the other reviewers.
Response 1: We have explicitly stated the limitations in the Discussion section, clarifying that the observed activation of caspase-3, 8, and 9, Furthermore, we have strengthened the Discussion by emphasizing the robustness of the SWATH-MS proteomic approach and the rigorous bioinformatic analyses employed to support our conclusions. Finally, we have added relevant bibliographic references to contextualize our methodological approach, citing studies where similar conclusions have been drawn based on gene expression or proteomics data alone.